# Adverse maternofoetal outcomes associated with ionised calcaemia, total calcaemia, albuminaemia, and calcium supplementation in pregnancy: Analysis from a resource-limited setting

Atem Bethel Ajong[1,2]☯*, Bruno Kenfack[3‡], Innocent Mbulli Ali[2‡], Martin Ndinakie Yakum[4‡], Prince Onydinma Ukaogo[5‡], Fulbert Nkwele Mangala[6,7‡], Loai Aljerf[8‡], Phelix Bruno Telefo[2]☯*

1 Department of Mother and Child Care, Kekem District Hospital, Kekem, West Region, Cameroon, 2 Department of Biochemistry, University of Dschang, Dschang, West Region, Cameroon, 3 Department of Obstetrics / Gynaecology and Maternal Health, Faculty of Medicine and Pharmaceutical Sciences, University of Dschang, Dschang, West Region, Cameroon, 4 Department of Epidemiology and Biostatistics, School of Medical and Health Sciences, Kesmonds International University, Bamenda, Cameroon, 5 Department of Pure and Industrial Chemistry, Abia State University, Uturu, Nigeria, 6 Faculty of Medicine and Pharmaceutical Sciences, University of Douala, Douala, Cameroon, 7 Maternity unit, Nkongsamba Regional Hospital, Nkongsamba, Littoral Region, Cameroon, 8 Faculty of Dentistry, Damascus University, Damascus, Syria

☯ These authors contributed equally to this work.
‡ BK, IMA, MNY, POU, FNM and LA also contributed equally to this work.
* christrah@yahoo.fr (ABA); bphelix@yahoo.co.uk (PBT)

## Abstract

### Introduction

Disorders of total calcium (tCa) in pregnancy have been associated with adverse materno-foetal outcomes. However, studies evaluating this from the viewpoint of ionised calcaemia are practically inexistent. This study estimates the prevalence of some adverse maternal and foetal outcomes and the potential effect of ionised calcium (iCa), tCa, albumin and calcium supplementation on some maternofoetal outcomes.

### Methods

A cross-sectional study was conducted among 1074 pregnant women in late pregnancy from four health facilities in the Nkongsamba Health District. Data were collected by interview, analysis of maternal blood samples and measurement of maternal and foetal parameters. Total calcaemia and albuminaemia were measured by atomic absorption spectrophotometry, while iCa and pH were measured using ion-selective potentiometry. Associations were measured using the odds ratio in simple and multiple logistic regression.

### Results

The prevalence of low birth weight, macrosomia, and hypertension in pregnancy was 6.27 [4.97–7.89]%, 4.78 [3.65–7.89]%, 10.24 [8.57–12.20]%, respectively. Following multiple

**Data Availability Statement:** All relevant data are within the article and its Supporting Information files.

**Funding:** The authors received no specific funding for this work.

**Competing interests:** The authors have declared that no competing interests exist.

logistic regression, women with iCa levels $\leq$ 1.31mmol/L had significantly increased odds of hypertension in pregnancy (AOR = 2.47 [1.63–3.74], p-value = 0.000), having babies with low birth weight (AOR = 2.02[1.33–3.61], p-value = 0.002), low birth length (AOR = 2.00 [1.34–2.99], p-value = 0.001), low brachial circumference (AOR = 1.41[1.10–1.81], p-value = 0.007), first minute Apgar score < 7 (AOR = 3.08[1.70–5.59], p-value = 0.000) and fifth minute Apgar score < 7 (AOR = 2.86[1.32–6.16], p-value = 0.007). Ionised calcaemia had no significant association with maternal body mass index immediately after birth and the head circumference of the baby. Total calcaemia was found to have no significant association with any of the selected outcomes, while women with total albuminaemia $\leq$ 30mg/L had significantly higher odds of having babies with low birth weight (AOR = 3.40[1.96–5.91], p-value = 0.000), and Apgar scores < 7 at the first (AOR = 2.07[1.16–3.70], p-value = 0.013). Calcium supplementation showed no significant association with any of the selected outcomes except for the first (OR = 0.42[0.24–0.72], p-value = 0.002) and fifth minute Apgar score (OR = 0.25[0.12–0.50], p-value = 0.000).

## Conclusion

The prevalence of low birth weight, macrosomia, and hypertension in pregnancy was 6.27 [4.97–7.89]%, 4.78 [3.65–7.89]%, 10.24 [8.57–12.20]%, respectively. Maternal iCa levels $\leq$ 1.31mmol/L significantly increase the odds of having babies with low birth weight, low birth length, low brachial circumference at birth, low Apgar scores at the first and fifth minutes and maternal hypertension in pregnancy. Low maternal albuminaemia is significantly associated with low birth weight, and Apgar score < 7 at the first minute. None f the selected maternofoetal outcomes directly depend on total calcaemia, given that none of the associations was statistically significant. Even though iCa levels remain relatively normal in normal pregnancies, it remains the strongest predictor of foetal outcomes. Calcium supplementation significantly improves the Apgar scores at the first and fifth minute. Routine pregnancy follow-up should include evaluating maternal calcaemic states, particularly the ionised fraction, to detect the low-normal concentrations likely to impact maternal and foetal outcomes. Normal iCa levels for pregnant women need revisiting.

## Introduction

Calcium in the human body contributes to about 1–2% of body weight, with 99% stored in the bone and teeth as hydroxyapatite [1–3]. The remaining 1% (5-6g) is found in the intracellular and extracellular milieu, with only 1.3g found in the extracellular milieu. In plasma (part of the extracellular milieu), calcium circulates in three principal forms. About 50% is free ionised calcium (iCa) and constitutes the metabolically active form of calcium, while 40% is bound and transported by plasma proteins (principally albumin). The remaining 10% is bound to small anions such as phosphate, carbonate, citrate, lactate, and sulphate [4, 5].

Hypocalcaemia in pregnancy is indisputably a widespread finding in pregnancy, particularly from the perspective of total calcaemia. In low and middle-income countries, the prevalence is very high, especially in the third trimester, where foetal calcium demands are highest [6]. The third-trimester prevalence varies from 58% in Cameroon [7], 70% in Algeria [8] and 66% in India [9]. According to literature, even though total calcium (tCa) levels vary and go

below the normal range in pregnancy, iCa levels are usually maintained within the normal range [6]. The measurement of iCa levels is complicated and liable to be influence by multiple factors. Moreover, the belief that iCa levels remain normal in pregnancy has discouraged research using ionised calcaemia in pregnancy and its routine use in clinical practice.

Low calcium intake significantly increases the likelihood of hypertensive diseases in pregnancy (gestational hypertension, pre-eclampsia and eclampsia) [10, 11], leads to low birth weight (LBW), low birth length (LBL) and small for gestational age babies [12]. In a recent study in India, total hypocalcaemia was significantly associated with LBW but had no statistically significant association with preterm delivery, pre-eclampsia and neonatal mortality [13]. Similar findings showing a non-statistically significant association between pre-eclampsia and total hypocalcaemia have been reported in other studies [14, 15]. The above findings remain controversial as contrasting evidence suggests a significant association between calcium levels and hypertensive diseases in pregnancy [16–18].

Literature evaluating the association between calcium supplementation in pregnancy and maternal and foetal outcomes like hypertensive diseases in pregnancy, and foetal birth weight (FBW) is available. However, studies that explore the relationship between total blood calcium levels (talk less of ionised calcaemia), and these outcomes are still sparse. In a very recent study where iCa was measured, ionised calcaemia was found to have a more robust prediction for pre-eclampsia than total calcaemia [19]. Most of these studies have been carried out around hypertensive diseases in pregnancy, and only a few have tried to verify which calcium fraction best predicts which outcomes. Moreover, studies comparing the effect of total calcaemia and ionised calcaemia on other foetal outcomes like BW, Apgar score (AS), brachial circumference (BC), and head circumference (HC) are practically inexistent.

Therefore, this write-up determines the potential effect of maternal blood calcium levels (tCa and iCa levels) and albumin on selected maternofoetal outcomes. It also presents the prevalence of some adverse obstetric outcomes and reassesses outcomes associated with calcium supplementation in pregnancy. This covers objective 3 of the registered research protocol previously published in PLoS one [20]. The manuscript evaluates maternofoetal hypocalcaemia-related outcomes, but given the strong relationship between albuminaemia and calcaemic states, as well as the link between calcium supplementation, calcaemic states and maternofoetal outcomes, the objectives of this manuscript were formulated as stated above. In this way, we will find out if each of the chosen factors had an association with the occurrence of adverse maternofoetal outcomes and also detect factors with a stronger association.

## Methodology

The methodology used in this research had been described and published in PLOS One before the onset of this study [20]. The specificities related to this write up have been described here.

### Study design and setting

This was a cross-sectional hospital-based study targeting apparently healthy pregnant women in late pregnancy (37 weeks of pregnancy and above) of the Nkongsamba Health District (NHD) between November 2020 and September 2021. The NHD is an extensive health district in the Moungo division, Littoral Region, Cameroon. Contrary to information presented in the original research protocol, four major health facilities of the NHD were considered for data collection in this study instead of the Nkongsamba Regional Hospital (NRH) alone [20]. These health facilities included the NRH, Bon Samaritain Medicalised Health Centre, Catholic Medicalised Health Centre, and the Fultang polyclinic. About 8 in 10 deliveries in this health district occur in these health facilities. These changes were made to have a more representative sample

of the NHD. Data were obtained by administering face-to-face, an interviewer-administered semi-structured questionnaire and additional data obtained through maternal biochemical blood assays.

## Eligibility, sample size, and sampling

Eligible participants were pregnant women at a gestational age of 37 weeks, and above, who were apparently healthy and were received at the maternities of the four included health facilities for antenatal care. Contrary to reports in the original protocol, which indicated that women with potential causes of hypocalcaemia were to be excluded, all apparently healthy pregnant women were included to have a general picture of the calcium profile in pregnant women of the health district. As presented in the protocol, we conducted this study with a calculated minimum sample size of 1067 participants, and sampling was exhaustive of all eligible and consenting pregnant women received during the study [20].

## Procedure of implementation, data collection and biochemical assays

As described in the protocol, after obtaining the required administrative authorisations, the data collection tools were pretested, and ethical clearance was obtained for the study. During three training sessions (5 hours each), seven midwives working in the selected maternities were trained on the study objectives, the consenting and the data collection procedure [20].

Data were collected on a one-to-one basis using a semi-structured interviewer-administered questionnaire. Following the World Health Organisation (WHO) best practice for phlebotomy [21], 10 ml each of blood was drawn into lithium heparinised and dry vacutainer tubes for the measurement of total calcaemia and ionised calcaemia, respectively. This blood collection was done on well-rested participants (10 minutes), seated with no fist formation and no prior physical exercise [22]. The samples were collected using vacuated needles with very short-lived use of tourniquets (less than 1 minute) [20].

Blood in heparinised tubes was used to measure tCa and albumin, while the blood collected in dry vacutainer tubes was settled (anaerobically) for 30 minutes and the serum extracted to measure iCa levels and pH. Total calcaemia and albuminaemia were measured using a semi-automatic KENZA MAX spectrophotometer (by atomic absorption spectrophotometry) following BIOLABO standard operating procedures. Total calcaemia was measured using O-Cresol Phtalein Complexone (CPC) [23], while total albumin measurement used Bromocresol green (BCG) [24] as reagents. Ionised calcaemia and pH were measured by ion-selective potentiometry using the K-lite 8 serum electrolyte analyser (MSLEA15-H model), with the manufacturer's standard operating procedures and reagents [20]. Measured iCa levels were then corrected for mild pH changes using the formula: Corrected $iCa^{2+}$ (pH 7.4) = Measured $iCa^{2+}$ [1−0.53 × (7.40−measured pH)] [25] which is valid between 7.2–7.6. Payne's equation was used to adjust tCa concentrations for albumin changes; $Ca_{adjusted}$(mmol/L) = tCa (mmol/L) + 0.02 [40 −albumin (g/L)] [26].

The blood pressure (BP) was taken in a seating position after 10 minutes of rest on the day of blood sample collection. Adapted cuffs and aneroid sphygmomanometers were used, with two BP measurements taken (separated by 2 minutes intervals) and the average BP calculated. The mean BP from the arm with the higher average was considered [27].

Second contact was made with participants at delivery for more data collection from the mother-baby pair. The mother's height and weight, foetal birthweight (FBW), birth length (BL), BC, HC, AS at the first and fifth minute were measured and registered in the questionnaires. Each participant's height was measured in an erect position, using a graduated height measuring scale (in meters). The weight and height of participants were measured just after

birth, with shoes off, light clothing, and empty pockets. Body Mass Index (BMI) was calculated as measured weight minus 1 kg (adjusting for clothing), divided by height squared (kg/m$^2$) [20]. The FBW was measured using a digital baby weighing scale and recorded to the nearest gramme in the questionnaire. The HC, BC, and BL were measured using an adapted measuring tape and recorded to the nearest centimetre in the questionnaire. The AS was evaluated using the five parameters of the APGAR score at the first and fifth minute following delivery.

## Data analysis

All questionnaires lacking essential information like ionised calcaemia, total calcaemia, and albuminemia were rejected. Data from retained questionnaires were entered into a predesigned data entry sheet on Epi-info, cleaned and analysed. Proportions and 95% confidence intervals (CIs) were estimated for categorical variables, while the means and standard deviations were used for continuous variables. Maternal and foetal variables (outcomes) were transformed into binary variables. LBW was defined as FBW less than 2500g [28], LBL as BL ≤ 48cm [29], low LBC as BC ≤ 11cm, and low AS as AS < 7 [30]. Hypertension in pregnancy was defined as Systolic BP ≥ 140mmHg and or diastolic BP ≥ 90mmHg [27]. To evaluate the prevalence of hypertensive diseases in pregnancy among hypocalcaemic women, a cut-off of 45mg/L and 85mg/L were used for ionised and total hypocalcaemia, respectively. Concerning the factors, ionised calcaemia was made binary using a cut-off of 1.31mmol/L (taken from the mean ionised calcaemia in the study), tCa was categorised using the cut-off of 85mg/L [20], and albuminaemia was categorised using the cut-off of 30mg/L (the mean concentration recorded in the study).

The association between the different factors (like iCa, tCa, albumin and calcium supplementation) and the selected outcomes (like FBW, BL, AS, and maternal hypertension) was measured using the odds ratio (OR) and its 95% CI. Factors with p-values <0.25, were included in the multiple logistic regression model with a statistical significance set at a p-value threshold of 0.05. The OR, Adjusted OR (AOR) and p-value were reported for each outcome.

## Ethics statement

Ethical clearance was obtained from the Cameroon Bioethics Initiative/ Ethics Review and Consultancy Committee CAMBIN/ ERCC in Yaounde, with **CBI/452/** ERCC/CAMBIN as the reference number. In addition, an information notice was prepared and was used to enlighten each participant on the risk and benefit of the study and what is expected of them. Consent was written and signed before enrollment. Young pregnant girls below 21years of age signed assent forms, and written consent was obtained from their parents, legal representatives or guardians. Participation was at will, and participants were free to leave the study without any penalisation. Only consenting and eligible participants were recruited for this study. All questionnaires were coded and remained confidential.

## Results

### Characteristics of the study population

A total of 1150 pregnant women in the late third trimester were contacted for this study. However, 76 eligible women refused to participate in the study, giving us a non-response rate of 6.61%. The ages of the participants ranged from 15 to 47 years, with a mean age of 28.20±6.08 years. The sampled population was dominated by pregnant women aged 21–30 years (56.52%), and about 9 in 10 (93.29%) had attended at least secondary school. A majority of

these women were cohabiting (39.89%), while 25.19% were legally married (65.09% in a union), and 34.74% single.

The mean total number of pregnancies was 3.46±2.06, while the average gestational age at the first antenatal visit(for the current pregnancy) was 17.10±8.08 weeks. About 2 in 10 women were in their first pregnancy, while most were in their second, third or fourth pregnancy (47.06%). About half (53.52%) initiated antenatal care after 4–6 months of pregnancy, while only about 1 in 10 women (11.62%) started antenatal care within the first two months of pregnancy. Concerning the number of antenatal visits, 87.14% had attended at least 4 antenatal care sessions.

## Prevalence of some adverse obstetric outcomes

Table 1 presents maternal and foetal outcomes. According to the results, the prevalence of hypertension in pregnancy was 10.24 [8.57–12.20]%. The prevalence of hypertension in pregnancy among women with hypocalcaemia was also evaluated (not on the table). This prevalence was 11.33 [9.13–13.97]% among women with total hypocalcaemia and 8.50 [6.17–11.59]% among women with total normocalcaemia (p-value = 0.136). As far as ionised hypocalcaemia is concerned, the prevalence of hypertension in pregnancy among women with ionised hypocalcaemia was 23.53 [10.75–41.17]% and 9.81 [8.15–11.77]% among women with ionised normocalcaemia (p-value = 0.009).

Concerning the FBW, 6.27 [4.97–7.89]% gave birth to babies with LBW, 23.78% had big babies at birth, while 4.78 [3.65–7.89]% had macrosomic babies (>4000g). Only 2.62 [1.82–3.76] % had babies with a fifth-minute AS < 7.

## Association of ionised calcaemia, total calcaemia and albuminaemia with selected maternofoetal outcomes

Table 2 shows the potential associations of calcaemic levels with some maternal and foetal outcomes. As far as LBW is concerned, ionised calcaemia and total albuminaemia havestatistically significant associations with LBW. Following multiple logistic regression, women with iCa levels ≤ 1.31 mmols/L had their odds of having babies with LBW increased by about 2 folds (AOR = 2.19[1.33–3.61], p-value = 0.002) while women with albumin levels ≤ 30 mg/L had their odds to deliver babies with LBW increased by up to 3.40 folds (AOR = 3.40[1.96–5.91], p-value = 0.000). Total calcaemia did not show any statistically significant association with LBW.

**Table 1. Prevalence of some adverse maternal and foetal outcomes.**

| Variable | Modalities | Proportion | Percentage |
|---|---|---|---|
| Hypertension in pregnancy (Systolic blood pressure≥140mmHg and/or Diastolic blood pressure ≥ 90mmHg) (n = 1074) | Yes | 110 | 10.24 |
| | No | 964 | 89.76 |
| Foetal birth weight (n = 1068) | <2500g | 67 | 06.27 |
| | 2500g≤BW<3500g | 696 | 65.17 |
| | 3500g≤BW<4000g | 254 | 23.78 |
| | ≥4000g | 51 | 04.78 |
| Apgar Score at first minute (n = 1068) | < 7 | 49 | 04.59 |
| | 8–10 | 1019 | 95.41 |
| Apgar Score at the fifth minute (n = 1070) | < 7 | 28 | 02.62 |
| | 8–10 | 1042 | 97.38 |

**Table 2. Association of ionised, total calcaemia and albuminaemia with maternofoetal outcomes.**

| Outcome | Factors in blood | Bivariate analysis | | Multivariate analysis | |
|---|---|---|---|---|---|
| | | OR [CI] | p-value | AOR [CI] | p-value |
| Low birth weight | Ionised calcium | 2.02 [1.25–3.28] | 0.004* | 2.19[1.33–3.61] | 0.002* |
| | Total Calcium | 1.47 [0.88–2.48] | 0.137 | 1.19[0.70–2.03] | 0.512 |
| | Albumin | 3.21[1.86–5.55] | 0.000* | 3.40[1.96–5.91] | 0.000* |
| | Albumin-corrected Calcium | 1.33[0.81–2.17] | 0.261 | | |
| First-minute Apgar score < 7 | Ionised calcium | 3.00[1.67–5.39] | 0.000* | 3.08[1.70–5.59] | 0.000* |
| | Total calcium | 1.55[0.85–2.81] | 0.150 | 0.86[0.23–3.23] | 0.824 |
| | Albumin | 1.91[1.08–3.38] | 0.026* | 2.07[1.16–3.70] | 0.013* |
| | Albumin-corrected calcium | 1.61[0.90–2.86] | 0.108 | 1.49[0.42–5.37] | 0.539 |
| Fifth-minute Apgar score < 7 | Ionised calcium | 2.88[1.35–6.15] | 0.006* | 2.86[1.32–6.16] | 0.007* |
| | Total Calcium | 1.90[0.85–4.27] | 0.120 | 1.64[0.36–7.44] | 0.523 |
| | Albumin | 1.65[0.80–3.42] | 0.175 | 1.73[0.83–3.61] | 0.144 |
| | Albumin-corrected calcium | 1.70 [0.78–3.63] | 0.168 | 0.95[0.23–3.91] | 0.943 |
| Hypertension in pregnancy | Ionised calcium | 2.55[1.68–3.84] | 0.000* | 2.47[1.63–3.74] | 0.000* |
| | Total Calcium | 1.38[0.90–2.10] | 0.138 | 0.88[0.34–2.30] | 0.797 |
| | Albumin | 0.86[0.58–1.28] | 0.467 | | |
| | Albumin-corrected calcium | 1.44[0.96–2.17] | 0.081 | 1.43[0.56–3.63] | 0.455 |
| Low birth length (≤48cm) | Ionised calcium | 2.04[1.37–3.04] | 0.001* | 2.00[1.34–2.99] | 0.001* |
| | Total calcium | 1.29[0.86–1.96] | 0.222 | 1.18[0.78–1.80] | 0.441 |
| | Albumin | 1.19[0.80–1.76] | 0.383 | | |
| | Albumin-corrected calcium | 1.11[0.74–1.65] | 0.616 | | |
| Brachial circumference ≤ 11cm | Ionised calcium | 1.43[1.11–1.84] | 0.005* | 1.41[1.10–1.81] | 0.007* |
| | Total calcium | 0.95[0.73–1.22] | 0.679 | | |
| | Albumin | 0.77[0.60–0.99] | 0.040* | 0.79[0.61–1.02] | 0.066 |
| | Albumin-corrected calcium | 1.03[0.80–1.33] | 0.799 | | |
| Head circumderence at birth ≤ 35cm | Ionised calcium | 0.80[0.63–1.03] | 0.080 | 0.82[0.64–1.05] | 0.111 |
| | Total calcium | 0.94[0.74–1.22] | 0.678 | | |
| | Albumin | 1.29[1.01–1.65] | 0.041* | 1.27[0.99–1.62] | 0.056 |
| | Albumin-corrected calcium | 0.99[0.77–1.26] | 0.923 | | |
| Maternal Body mass index after birth above 25kg/m$^2$ | Ionised calcium | 1.02[0.79–1.33] | 0.869 | | |
| | Total calcium | 0.99[0.76–1.29] | 0.936 | | |
| | Albumin | 1.03[0.80–1.34] | 0.807 | | |
| | Albumin-corrected calcium | 0.97[0.74–1.26] | 0.802 | | |

*Statistically significant, OR = Odds Ratio, AOR = Adjusted OR, CI = Confidence Interval

Concerning the first minute AS, ionised calcaemia and albuminaemia were found to have a statistically significant association with the AS at the first minute. Following multiple logistic regression, women who had iCa levels ≤ 1.31 mmol/L had their odds of having babies with first-minute AS < 7 increased by 3.08 folds (AOR = 3.08 [1.70–5.59], p-value = 0.000), while women with albumin levels ≤ 30mg/L had their odds of having babies with the first-minute AS < 7 increased by 2.07 folds (AOR = 2.07[1.16–3.70], p-value = 0.013). Concerning the fifth-minute AS, only ionised calcaemia had an effect. Following multiple logistic regression, women with iCa levels ≤1.31 mmol/L had their odds of having babies with a fifth-minute AS < 7 increased by 2.86 folds (AOR = 2.86[1.32–6.16], p-value = 0.007). Total calcaemia had no statistically significant association with AS.

Concerning hypertension in pregnancy, only ionised calcaemia had a statistically significant association with hypertension in pregnancy. Following multiple logistic regression, women with iCa levels ≤ 1.31mmol/L had their odds of hypertension in pregnancy increased by 2.47 folds (AOR = 2.47[1.63–3.74], p-value = 0.000). Total calcaemia and albuminaemia did not significantly affect the odds of hypertension in pregnancy.

**Table 3. Association of calcium supplementation in pregnancy with maternal and foetal outcomes.**

| Outcomes | Odds Ratio | p-value |
|---|---|---|
| Maternal body mass index immediately after birth above 25 kg/m$^2$ | 1.16[0.87–1.56] | 0.296 |
| Hypertension in pregnancy | 0.91[0.59–1.41] | 0.672 |
| Low birth weight (< 2500g) | 0.71[0.43–1.17] | 0.174 |
| Low birth length (≤ 48cm) | 0.89[0.58–1.39] | 0.626 |
| Brachial circumference ≤ 11cm | 1.22[0.92–1.63] | 0.160 |
| Head circumference ≤ 35cm | 1.05 [0.80–1.39] | 0.704 |
| Apgar score at first minute < 7 | 0.42 [0.24–0.72] | 0.002* |
| Apgar score at fifth minute < 7 | 0.25 [0.12–0.50] | 0.000* |

*Statistically significant

Concerning the BL, only ionised calcaemia it was found to have an effect. Following multiple logistic regression, women with iCa levels ≤ 1.31mmol/L had their odds of LBL in the babies increased by 2 folds (AOR = 2.00[1.34–2.99], p-value = 0.001). Total calcaemia and albuminaemia did not significantly affect BL. Concerning the BC, only ionised calcaemia was found to have an impact. Women with iCa levels ≤ 1.31mmol/L had their odds of LBC (≤11cm) in the baby increased by 1.41 folds (AOR = 1.41[1.10–1.81], p-value = 0.007) compared to those with higher ionised calcaemia. Total calcaemia and albuminaemia were not found to have any effect. Concerning the HC and maternal BMI immediately after birth, none of the factors had a statistically significant relationship with any of the three factors.

Table 3 shows the outcomes associated with calcium supplementation in pregnancy. Calcium supplementation was found to have non-statistically significant associations with maternal BMI immediately after birth, hypertension in pregnancy, LBW, LBL, LBC, and HC. However, calcium supplementation had a statistically significant association with the AS at the first and fifth minutes. Women who took calcium supplements in pregnancy had their odds of having babies with the first-minute AS < 7 reduced by 0.42 folds (OR = 0.42[0.24–0.72], p-value = 0.002). Also, women who took calcium supplements in pregnancy had their odds of having babies with a fifth-minute AS < 7 reduced by 0.31 folds (AOR = 0.25[0.12–0.50], p-value = 0.000).

## Discussion

This study evaluates the relative effect of total calcaemia, ionised calcaemia, albuminaemia and calcium supplementation on some key obstetric outcomes. The prevalence of LBW, macrosomia, and hypertension in pregnancy was 6.27 [4.97–7.89]%, 4.78 [3.65–7.89]%, 10.24 [8.57–12.20]%, respectively. Following multiple logistic regression, women with iCa levels ≤ 1.31mmol/L had significantly increased odds of hypertension in pregnancy (AOR = 2.47 [1.63–3.74], p-value = 0.000), having babies with LBW (AOR = 2.02[1.33–3.61], p-value = 0.002), LBL (AOR = 2.00 [1.34–2.99], p-value = 0.001), LBC (AOR = 1.41[1.10–1.81], p-value = 0.007), first minute AS < 7 (AOR = 3.08[1.70–5.59], p-value = 0.000) and fifth minute AS < 7 (AOR = 2.86[1.32–6.16], p-value = 0.007). Ionised calcaemia was not significantlt associated with maternal BMI immediately after birth and the HC of the baby. Total calcaemia was found to have no association with any of the selected outcomes while women with albuminemia ≤ 30mg/L had increased odds of having LBW babies (AOR = 3.40[1.96–5.91], p-value = 0.000), and babies with AS ≤ 7 at the first minute (AOR = 2.07[1.16–3.70], p-value = 0.013). Calcium supplementation was not associated with any of the selected outcomes

except for the first (OR = 0.42[0.24–0.72], p-value = 0.002) and fifth minute AS (OR = 0.25 [0.12–0.50], p-value = 0.000).

According to the WHO, a baby has LBW if born with a weight below 2500g, irrespective of the gestational age [31, 32]. LBW is responsible for 60–80% of neonatal deaths in developing countries [28]. Our study included women at term (at least 37weeks complete weeks) and had a prevalence of LBW at term of 6.27%. As expected, this is smaller than the prevalence of 13.5% reported in Buea and the 11% national prevalence [28], which considered deliveries irrespective of gestational age. The prevalence in this study corresponds more to the proportion of babies who might have experienced intra-uterine growth restriction or, better still, were just small for their gestational age babies at term [33, 34].

On the other hand, the prevalence of macrosomia was 4.78 [3.65–7.89]%, which is lower than the 6.41% reported in Cameroon about two decades ago [35]. This 6.41% is, however, found in our reported CI. This is also smaller than the findings in a recent study in Buea, which reported a prevalence of macrosomia of 9.5% [36]. These discrepancies may be due to changes in nutritional behaviours in pregnancy and the evolving quality of antenatal care over the years. Macrosomia has been associated in literature with labour dystocia, increased likelihood of severe perineal tears, higher caesarean section rate and instrumental delivery, postpartum haemorrhage, poor Apgar scores at the fifth minute and neonatal death [37–39].

According to our results, 10.24% had hypertensive disorders in pregnancy. Hypertensive disorder in pregnancy is a generic term that includes pre-existing and gestational hypertension, pre-eclampsia and pre-eclampsia with convulsions (eclampsia) [40, 41]. Hypertensive disorders in pregnancy remain a significant cause of maternal and foetal morbi-mortality [42], standing out as the second leading cause of maternal mortality worldwide [43]. The prevalence of hypertensive diseases in pregnancy recorded in this study is similar to the global prevalence of 10% [44].

Several studies have evaluated the relationship between maternal blood calcium levels and hypertension in pregnancy. However, studies that considered the effect of iCa are rare. Following multiple logistic regression, women with iCa levels $\leq$ 1.31mmol/L had their odds of hypertension in pregnancy increased by 2.47 folds (AOR = 2.47[1.63–3.74], p-value = 0.000). In a study in which tCa was used, the mean total calcaemia among women with pregnancy-induced hypertension was significantly lower than that in women with normal pregnancy [16]. Also, in a very recent Ethiopian study, similar findings have been reported, and in addition, preeclampsia was found to have a stronger association with iCa compared with tCa [19]. However, contrary to reports in this recent study, we did not find total calcaemia to influence the likelihood of hypertension in pregnancy. Our results suggest that calcium intake and supplementation in regulating blood pressure in pregnancy do so by affecting the ionised fraction of calcium in blood. According to the pathophysiology, low calcium levels induce the production of the parathyroid hormone, which in turn leads to increased intracellular calcium concentrations and subsequent vasoconstriction, thereby increasing blood pressure [19]. This study stands out as one of the pioneer contributions that evaluate the relative impact of ionised and tCa on blood pressure values in pregnancy. The recent study carried out in Ethiopia was focused only on pre-eclampsia, while this study considers hypertension in pregnancy as a block.

In this study, calcium supplementation was not found to affect ionised and tCa levels, even though tCa levels were positively associated with iCa levels (r = 26.12, p-value = 0.000). The same non-significant relationship between calcium intake and tCa has been reported in India [14] and Italy [45]. Similar findings have been found in a recent Ethiopian study that evaluated both total and ionised calcaemia [19]. These results suggest that calcium supplementation might not be associated with lower blood pressure values in pregnancy or contributes to

reducing blood pressure not by increasing total calcaemia, but by maintaining a stable concentration of ionised calcaemia.

Multiple studies have reported that total calcaemia in pregnancy has no effect on FBW [7, 46, 47]. In line with these findings, our results found no association between total calcaemia and FBW. However, we found ionised calcaemia to be a strong potential factor that affects FBW. Following multiple logistic regression, women with iCa levels ≤ 1.31 mmols/L had their odds of having babies with LBW increased by about 2 folds (AOR = 2.19[1.33–3.61], p-value = 0.002). Our findings were difficult to compare, given that this was practically the first evaluation of this association. This can be explained by the inevitable intervention of calcium signalling in the different growth and differentiation pathways [48]. In addition, better iCa concentrations in maternal blood will mean more calcium available to cross to the baby for bone development. Calcium supplementation did not show a statistically significant association with FBW. Likely, calcium supplementation does not affect FBW. This has been reported by other studies and systematic reviews [49, 50]. However, contrasting evidence has suggested beneficial effects of increased calcium intake in reducing the likelihood of LBW [51, 52].

Serum albumin levels have been described as a marker protein for nutritional status [53]. In our study, women with albumin levels ≤ 30 mg/L had their odds of having babies with LBW increased by 3.40 folds (AOR = 3.40[1.96–5.91], p-value = 0.000). Here, low albumin is likely not to be a direct cause of LBW but might be a marker of the limited availability of amino acids for the synthesis of different structural proteins of the baby, hence the effect on FBW. Similar results have been found in Japan [54] and Nigeria [53].

Moreover, significant associations were found between iCa levels and other parameters like the BL and BC. Following multiple logistic regression, women with iCa levels ≤ 1.31mmol/L had their odds of LBL in the babies increased by 2 folds (AOR = 2.00[1.34–2.99], p-value = 0.001). Also, women with iCa levels ≤ 1.31mmol/L had their odds of LBC (≤11cm) in the baby increased by 1.41 folds (AOR = 1.41[1.10–1.81], p-value = 0.007). To the best of our knowledge, no studies have evaluated this relationship and compared the effect of ionised and tCa on these outcomes. There is, therefore, no data for comparison. These observations can be explained by the interference of available iCa in signalling pathways and the consequential availability of diffusible calcium for foetal bone development. These outcomes are dependent on iCa and not the tCa. In a prospective cohort study, high parathyroid hormone with low 25 (OH)D or very little calcium intake was significantly associated with LBL and lower HC [55]. However, the direct influence of calcium and the different fractions was not evaluated in this study.

Our study associated higher ionised serum calcium levels with better foetal outcomes in terms of the AS at the first and fifth minute. We did not find any studies evaluating the effect of iCa levels on the AS and therefore could not compare our results. However, better concentrations of the physiologically active fraction of calcium should be associated with a better physiological response of the baby at birth. A nutritional state marker like albumin was found to significantly predict the AS at birth. Lower albumin concentrations could mean limited protein nutrients and amino acids required for the normal foetal response at birth. Albumin might not be directly involved but could be an explorable marker of maternal nutritional state and, therefore, a good predictor of foetal response at birth. Calcium supplementation could directly improve the ionised calcaemic poll required for the response at birth. This explains the association observed between calcium supplementation in pregnancy and AS. Similar results have been found in literature showing that calcium supplementation in pregnancy is significantly associated with a reduced likelihood of having babies with fifth-minute AS < 7 [56].

## Limits of the study

The associations established in this write up should be taken with care. The cross-sectional nature of this research only allows for the emission of hypotheses and not causal relationships. Calcium concentrations were measured only once, and therefore acute variations could not be differentiated from chronic variations. Moreover, our study evaluated outcomes only at term and birth. Even though adverse maternofoetal outcomes associated with hypocalcaemia are expected to have effects on neonatal health, our study did not assess the effect of maternal calcaemic states on neonatal health at key periods of neonatal follow-up. Notwithstanding, the relationships established in this manuscript are pertinent, original, even if they require further investigations with more adapted study designs. Our research stands out as a pioneer study in evaluating the effect of iCa, tCa, and albumin on foetal outcomes.

## Conclusion

The prevalence of LBW, macrosomia and hypertension in pregnancy was 6.27 [4.97–7.89]%, 4.78 [3.65–7.89]%, 10.24 [8.57–12.20]%, respectively. Maternal iCa levels $\leq$ 1.31mmol/L significantly increase the odds of having babies with LBW, LBL, LBC, low AS at the first and fifth minute and maternal hypertension in pregnancy. Low maternal albuminemia is significantly associated with LBW and AS < 7 at the first minute. Maternofoetal outcomes do not depend on total calcaemia, given that it does not affect BW, hypertension in pregnancy, AS, BL, BC, and BMI immediately after birth.

Even though iCa levels remain relatively normal in normal pregnancies, it remains the strongest predictor of foetal outcomes. Calcium supplementation in pregnancy does not significantly affect these outcomes, but the direction of influence ties with observations of calcaemic effects. Notwithstanding, calcium supplementation in pregnancy significantly affects the AS at the first and fifth minutes. Routine pregnancy follow-up should include routine evaluation of maternal calcaemic states, particularly the ionised fraction, to detect the low-normal concentrations likely to impact maternal and foetal outcomes. The effect of ionised calcaemia on maternofoetal outcomes suggests a revisiting of the normal cut-offs for ionised calcaemia in pregnancy.

## Supporting information

**S1 Data. Data base of determinants and effects of low serum calcium in pregnancy.** (ACCDB)

## Acknowledgments

Our sincere gratitude goes out to:

- The Director of the NRH and Bethanie Group of laboratories for their support,

- Fouko Eric Dagobert and collaborators for their assistance in the laboratory,

- Midwives who participated in data collection,

- Matcha Waffo Lea Patricia for her contribution to data entry, and

- Pregnant women who consented to participate in this study.

Some changes have occurred on the author list compared with the registered protocol published.

## Author Contributions

**Conceptualization:** Atem Bethel Ajong, Bruno Kenfack, Innocent Mbulli Ali, Martin Ndinakie Yakum, Prince Onydinma Ukaogo, Loai Aljerf, Phelix Bruno Telefo.

**Data curation:** Atem Bethel Ajong, Martin Ndinakie Yakum, Fulbert Nkwele Mangala, Loai Aljerf, Phelix Bruno Telefo.

**Formal analysis:** Atem Bethel Ajong, Martin Ndinakie Yakum, Phelix Bruno Telefo.

**Funding acquisition:** Atem Bethel Ajong.

**Investigation:** Atem Bethel Ajong, Bruno Kenfack, Innocent Mbulli Ali, Fulbert Nkwele Mangala.

**Methodology:** Atem Bethel Ajong, Bruno Kenfack, Innocent Mbulli Ali, Martin Ndinakie Yakum, Prince Onydinma Ukaogo, Fulbert Nkwele Mangala, Loai Aljerf, Phelix Bruno Telefo.

**Project administration:** Atem Bethel Ajong, Phelix Bruno Telefo.

**Resources:** Atem Bethel Ajong.

**Software:** Atem Bethel Ajong, Prince Onydinma Ukaogo, Loai Aljerf.

**Supervision:** Atem Bethel Ajong, Bruno Kenfack, Innocent Mbulli Ali, Fulbert Nkwele Mangala, Loai Aljerf, Phelix Bruno Telefo.

**Validation:** Atem Bethel Ajong, Bruno Kenfack, Innocent Mbulli Ali, Martin Ndinakie Yakum, Prince Onydinma Ukaogo, Fulbert Nkwele Mangala, Loai Aljerf.

**Visualization:** Atem Bethel Ajong, Bruno Kenfack, Innocent Mbulli Ali, Phelix Bruno Telefo.

**Writing – original draft:** Atem Bethel Ajong.

**Writing – review & editing:** Atem Bethel Ajong, Bruno Kenfack, Innocent Mbulli Ali, Martin Ndinakie Yakum, Prince Onydinma Ukaogo, Fulbert Nkwele Mangala, Loai Aljerf, Phelix Bruno Telefo.

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
