## [Decision Letter · Decision Letter 0]

21 Jan 2022

PONE-D-21-33966How does ionised calcium, total calcium, total albumin, and calcium supplementation affect maternofoetal outcomes? analysis from a resource-limited settingPLOS ONE

Dear Dr. Ajong,

Thank you for submitting your manuscript to PLOS ONE. After careful consideration, we feel that it has merit but does not fully meet PLOS ONE’s publication criteria as it currently stands. Therefore, we invite you to submit a revised version of the manuscript that addresses the points raised during the review process.

We look forward to receiving your revised manuscript.

Kind regards,

Linglin Xie

Academic Editor

PLOS ONE

Journal Requirements:

Reviewers' comments:

Reviewer's Responses to Questions

**Comments to the Author**

1. Does the manuscript adhere to the experimental procedures and analyses described in the Registered Report Protocol?

If the manuscript reports any deviations from the planned experimental procedures and analyses, those must be reasonable and adequately justified.

Reviewer #1: Yes

2. If the manuscript reports exploratory analyses or experimental procedures not outlined in the original Registered Report Protocol, are these reasonable, justified and methodologically sound?

A Registered Report may include valid exploratory analyses not previously outlined in the Registered Report Protocol, as long as they are described as such.

Reviewer #1: Yes

3. Are the conclusions supported by the data and do they address the research question presented in the Registered Report Protocol?

The manuscript must describe a technically sound piece of scientific research with data that supports the conclusions. The conclusions must be drawn appropriately based on the research question(s) outlined in the Registered Report Protocol and on the data presented.

Reviewer #1: Yes

4. Have the authors made all data underlying the findings in their manuscript fully available?

Reviewer #1: No

5. Is the manuscript presented in an intelligible fashion and written in standard English?

Reviewer #1: No

6. Review Comments to the Author

Please use the space provided to explain your answers to the questions above. (Please upload your review as an attachment if it exceeds 20,000 characters)

Reviewer #1: Review comments on PONE-D-21-33966

Major comments

This is an article discussing an association between pregnant maternal serum calcium, ionized calcium and fetal outcomes including anthropometric indecies and physiological status using Apgar scores (AS) at on and five minutes after birth. The authors successfully showed an association of ionized calcium and low birth weight, low AS scores, and brachial circumferences among 1,074 pregnant women. However, the reason why the authors pay attention to serum calcium and fetal growth and no information how amount of calcium supplement or calcium daily intake had during each trimester as the authors discussed in the discussion part. Moreover, they did not show mother’s nutritional status including BMI at baseline and body weight gain during pregnancy. Unless such information is not shown, clinicians could not identify who is at risk of low ionized calcium concentration from mother’s nutrition. Additionally, English writing seems poor to be edited by native English writers. The timing to take blood also must be shown whether before breakfast, fating period, or early morning to standardize serum calcium concentrations.

In conclusion, I feel that this article must show mother’s nutritional status and daily nutrients intake in whom serum ionized calcium is lower than cutoff value and English edits. However, the authors’ eye point of ionized calcium with potential function of fetal cell development seems opening novel scientific door to faeto-maternal affect. As my conclusion, I must ask major revision for the points above mentioned. Otherwise, this must be rejected.

Minor comments

1. No explanations of abbreviations in all tables.

2. Why they measured brachial circumference? I feel this must be overestimated because of effect of long-term floating in yolk sac fluid before birth.

3. What contents of interview questionnaire and was it validated, although the authors described that questionnaires were validated but no references to explain that?

4. Line 212, “was” was duplicated in a single sentence.

5. The part of the end of discussion must be divided to “Limitations of the study” like the other articles.

6. Line 387: the word of “cause-effect relationship” might be changed to causal association.

7. PLOS authors have the option to publish the peer review history of their article (what does this mean?). If published, this will include your full peer review and any attached files.

Reviewer #1: **Yes: **Teruyoshi Amagai, MD, PhD

---

## [Author Response · Author response to Decision Letter 0]

28 Jan 2022

Dear Editor/Reviewer,

We wish to thank you for the contributions you have been put in to strengthen this manuscript. We have taken time and gone through each point and have responded while making changes to the manuscript. The response to reviewers comments have been presented here in a tabular form (below). Also, the recommendations of the editor in terms of PLoS one guidelines, ethical statement and data availability statement have been addressed.

REVIEWER’S COMMENTS 

This is an article discussing an association between pregnant maternal serum calcium, ionized calcium and fetal outcomes including anthropometric indecies and physiological status using Apgar scores (AS) at on and five minutes after birth. The authors successfully showed an association of ionized calcium and low birth weight, low AS scores, and brachial circumferences among 1,074 pregnant women. 

Resp: Thank you for your observation.

However, the reason why the authors pay attention to serum calcium and fetal growth and no information how amount of calcium supplement or calcium daily intake had during each trimester as the authors discussed in the discussion part. 

Resp: Thanks for this observation. We designed and developed a research protocol which we submitted as was published in PLoS One (1). The present manuscript provides a report of some of the research findings. While we do acknowledge the limits associated with the cross sectional design (stated in the limits), we seek to faithfully report the outcomes of the study as designed in the protocol and recommended by PLoS One editorial policies on open science. This work has been divided into three major write ups which have all been submitted to PLoS One.

We have a manuscript evaluating calcium intake (in terms of calcium supplementation practices in pregnancy) and determines factors affecting this practice in pregnancy (given that it is recommended by the WHO to be systematic in LMIC). This was submitted on the 1st 0ct 2021 (PONE-D-21-31679).

The second manuscript from this registered protocol focused on evaluating the prevalence of ionised and total hypocalcaemia and associated socio-demographic, obstetric and nutritional factors using standard statistics. It also finds out how calcium supplementation variables like daily dose, duration, taking calcium with other supplements affect calcaemic states. This has already gone through revision (PONE-D-21-31669R1)

The present manuscript therefore targets the hypocalcaemia-associated outcomes. It evaluates the prevalence of some adverse maternofoetal outcomes and finds out associations between calcaemic states (ionised and total calcium), albumin and maternofoetal outcomes. We feel that mixing calcium intake/calcium supplementation into this manuscript does not help us respond to any of its objectives. This aspect has been dealt with in our first manuscript.

Moreover, they did not show mother’s nutritional status including BMI at baseline and body weight gain during pregnancy. Unless such information is not shown, clinicians could not identify who is at risk of low ionized calcium concentration from mother’s nutrition. Resp: We thank the Reviewer for this point. As stated in the in the registered protocol, we were interested in evaluating nutritional habits or behaviours and how they influence calcaemic states (already addressed in the second manuscript). We acknowledge that the nutritional status of these women was not evaluated in our protocol. 

As stated in the registered protocol, our study was cross sectional in design and did not aim at finding causative factors of low hypocalcaemia in pregnancy (which would ideally be done using a longitudinal design). Evaluation of risk factors associated with hypocalcaemia is addressed by the second manuscript.

We used a crude BMI which was gotten immediately after delivery. Measurement of overweight and obesity in pregnancy using BMI was not one of our objectives. We believed that the error associated with its evaluation immediately after birth was common in all included women and the trend could therefore still be useful.

We understand that criticisms of this sort greatly improves the study and lends another dimension of its utility. We have taken note of this and will incorporate the pertinent points in subsequent research on this important subject

Additionally, English writing seems poor to be edited by native English writers. 

Resp: Thanks for your observation. The whole manuscript has been edited by a native English speaker.

The timing to take blood also must be shown whether before breakfast, fating period, or early morning to standardize serum calcium concentrations. 

Resp: Thanks for your pertinent observation. Blood sample collection time for calcium assays was random and depended on the time at which the participant was included into the study as has been the case in similar studies (2–6). No particular fasting indications were given to participants before sample collection. However, as stated in the methods section, precautions were taken to make sure measured calcium levels best reflect the calcaemic states of the participants.

In conclusion, I feel that this article must show mother’s nutritional status and daily nutrients intake in whom serum ionized calcium is lower than cutoff value and English edits. 

Resp: We want to thank the Reviewer for the time taken to assess the value of this study, which he greatly appreciates and therefore provided constructive feedback. We have taken note of some of the pertinent comments which, unfortunately, can only be adequately addressed in a new study (longitudinal). We plan to take these into consideration as part of our future study. 

However, the authors’ eye point of ionized calcium with potential function of fetal cell development seems opening novel scientific door to faeto-maternal affect. 

Resp: Thank you very much.

As my conclusion, I must ask major revision for the points above mentioned. Otherwise, this must be rejected. 

Resp: Thanks for your recommendation. We have provided a point by point response to the comments and substantially revised the manuscript. We remain open should there be other issues to address prior to publication of the manuscript following the publication of its protocol.

MINOR REVIEWER’S COMMENTS RESPONSES

1. No explanations of abbreviations in all tables.

 Resp: Thanks for your observations, the tables have been edited and all abbreviations removed. See table 1, 2 and 3.

2. Why they measured brachial circumference? I feel this must be overestimated because of effect of long-term floating in yolk sac fluid before birth.

 Resp: Thanks for the pertinent point. We agree with the Reviewer that this measurement might be overestimated. But our aim was not to define with exactitude the prevalence of low brachial circumference at birth. In addition, we needed the brachial circumference at birth which is standardly measured like that in the hospitals (7,8). Moreover, even with the error, the trend of variation with calcaemic states will still be detected (7,8).

3. What contents of interview questionnaire and was it validated, although the authors described that questionnaires were validated but no references to explain that?

 Resp: Thanks for your observations. The questionnaire used in this study was published with the registered protocol which has been cited in the manuscript. The sentence fraction “validated questionnaires” is only used in the first sentence of the data analysis section and describes retained questionnaires. To eliminate the ambiguity, the word has been replaced with “retained”. See page 9, paragraph 1.

4. Line 212, “was” was duplicated in a single sentence.

 Resp: Thanks for the observation. The error has been corrected. See line 217, paragraph 2, page 11.

5. The part of the end of discussion must be divided to “Limitations of the study” like the other articles.

 Resp: Thanks for your recommendation. The section “limits of the study” has been added. See paragraph 1, page 21

6. Line 387: the word of “cause-effect relationship” might be changed to causal association. 

 Resp: Thanks for your suggestion. It has been changed to causal. See paragraph 1, page 21

 

References

1. Ajong AB, Kenfack B, Ali IM, Yakum MN, Aljerf L, Telefo PB. Hypocalcaemia and calcium intake in pregnancy: A research protocol for critical analysis of risk factors, maternofoetal outcomes and evaluation of diagnostic methods in a third-category health facility, Cameroon. PLoS One. 2020;15(11 November). 

2. Ajong AB, Kenfack B, Ali IM, Yakum MN, Telefo PB. Prevalence and correlates of low serum calcium in late pregnancy: A cross sectional study in the Nkongsamba Regional Hospital; Littoral Region of Cameroon. PLoS One. 2019;14(11). 

3. Bako B, El-Nafaty AU, Mshelia DS, Gali RM, Isa B, Dungus MM. Prevalence and risk factors of hypocalcemia among pregnant and non-pregnant women in Maiduguri, Nigeria: A cross-sectional study. Niger J Clin Pract. 2021; 

4. Gebreyohannes RD, Abdella A, Ayele W, Eke AC. Association of dietary calcium intake, total and ionized serum calcium levels with preeclampsia in Ethiopia. BMC Pregnancy Childbirth. 2021;21(1). 

5. Kumar A, Agarwal K, Devi SG, Gupta RK, Batra S. Hypocalcemia in Pregnant Women. Biol Trace Elem Res. 2010 Jul;136(1):26–32. 

6. Benali AI, Demmouche A. Calcium Deficiency among Pregnant Women and their Newborns in Sidi Bel Abbes Region, Algeria. J Nutr Food Sci. 2014;04(06):4–7. 

7. Shankar R, Ramarajan A, Rani S, Seshiah V. Anthropometric and Skin Fold Thickness Measurements of Newborns of Gestational Glucose Intolerant Mothers: Does it Indicate Disproportionate Fetal Growth? J Obstet Gynecol India. 2020; 

8. Tiruneh C, Teshome D. Prediction of Birth Weight by Using Neonatal Anthropometric Parameters at Birth in Finote Selam Hospital, Ethiopia. Pediatr Heal Med Ther. 2021;

---

## [Decision Letter · Decision Letter 1]

4 Apr 2022

PONE-D-21-33966R1How does ionised calcium, total calcium, total albumin, and calcium supplementation affect maternofoetal outcomes? analysis from a resource-limited settingPLOS ONE

Dear Dr. ajong,

Thank you for submitting your manuscript to PLOS ONE. After careful consideration, we feel that it has merit but does not fully meet PLOS ONE’s publication criteria as it currently stands. Therefore, we invite you to submit a revised version of the manuscript that addresses the points raised during the review process. Especially, please address the comments from reviewer #1.

We look forward to receiving your revised manuscript.

Kind regards,

Linglin Xie

Academic Editor

PLOS ONE

Reviewers' comments:

Reviewer's Responses to Questions

**Comments to the Author**

1. Does the manuscript adhere to the experimental procedures and analyses described in the Registered Report Protocol?

If the manuscript reports any deviations from the planned experimental procedures and analyses, those must be reasonable and adequately justified.

Reviewer #1: No

Reviewer #2: Yes

2. If the manuscript reports exploratory analyses or experimental procedures not outlined in the original Registered Report Protocol, are these reasonable, justified and methodologically sound?

A Registered Report may include valid exploratory analyses not previously outlined in the Registered Report Protocol, as long as they are described as such.

Reviewer #1: Partly

Reviewer #2: Yes

3. Are the conclusions supported by the data and do they address the research question presented in the Registered Report Protocol?

The manuscript must describe a technically sound piece of scientific research with data that supports the conclusions. The conclusions must be drawn appropriately based on the research question(s) outlined in the Registered Report Protocol and on the data presented.

Reviewer #1: No

Reviewer #2: Yes

4. Have the authors made all data underlying the findings in their manuscript fully available?

Reviewer #1: No

Reviewer #2: Yes

5. Is the manuscript presented in an intelligible fashion and written in standard English?

Reviewer #1: No

Reviewer #2: Yes

6. Review Comments to the Author

Please use the space provided to explain your answers to the questions above. (Please upload your review as an attachment if it exceeds 20,000 characters)

Reviewer #1: Reviewer’s comments on PLOE-D-21-33966 R1

After the authors revised an article entitled:

An article seems more sophisticated and scientific. However, its devised version does not change the results and their clinical meanings to improve outcomes on maternal and fetal health. The authors found that ionized Ca of the pregnant mothers seems to have impacts on fetal outcomes, including Apgar scores at one and five minutes after birth. However, the issues of neonates must be occurrence of events seen at longer observational periods of time, such as 7-day, 14-day, or 28-day mortality or IRDS. If neonates have lower Apgar score (AS) at birth, neonatologists could treat them properly and AS seems no effects on 7-, 14-,28-days mortality.

In conclusion, although the authors had tremendously a lot of time to study, the results seems no clinical impacts on neonates and this might be rejected.

Reviewer #2: All comment raised by the first and second revision are addressed:

Thanks for that and congratulation.

7. PLOS authors have the option to publish the peer review history of their article (what does this mean?). If published, this will include your full peer review and any attached files.

Reviewer #1: No

Reviewer #2: No

---

## [Author Response · Author response to Decision Letter 1]

9 Apr 2022

RESPONSE TO REVIEWER’S COMMENTS

We wish to appreciate once more the efforts of the editor and the reviewers to improve the quality of this manuscript. We have presented the response to each reviewer’s comment in the table below.

Comments and Response

1. Does the manuscript adhere to the experimental procedures and analyses described in the Registered Report Protocol?

If the manuscript reports any deviations from the planned experimental procedures and analyses, those must be reasonable and adequately justified.

Reviewer #1: No

Reviewer #2: Yes

 Resp: Thank you for your response. Our manuscript partly adheres to experimental procedures and analyses reported in the registered protocol. All the data collection steps were respected from the number of participants, recruitment of participants, methods of biochemical assays, measurement of maternal foetal parameters and data analysis. No additional data were collected to be able to tackle the objectives set in this manuscript.

We reported a slight deviation in the methods where we explained why instead of considering only the Nkongsamba Regional Hospital, we had to include 4 major health facilities in the Nkongsamba Health District. This is explained in the manuscript with the reason. Also, instead of excluding a set of participants with hypocalcaemia-causing pathologies, we including all women who were apparently healthy in the study and this statement was included and justified. See methods,line 121-132 and line 138-141. Page 8.

2. If the manuscript reports exploratory analyses or experimental procedures not outlined in the original Registered Report Protocol, are these reasonable, justified and methodologically sound?

A Registered Report may include valid exploratory analyses not previously outlined in the Registered Report Protocol, as long as they are described as such.

Reviewer #1: Partly

Reviewer #2: Yes

Resp: Thank you for your response. This manuscript was written out from one of the objectives of the registered protocol. 

Objective 3: Determine the prevalence of hypertensive disorders in pregnancy among hypocalcaemic pregnant women in the NRH and describe hypocalcaemia-associated maternofoetal outcomes (with foetal outcomes evaluated assecondary outcomes) (1). We reformulated this objective to that described in the manuscript because of the links between albumin, total and ionised calcaemia. We were interested in finding out if associations existed between these factors and maternofoetal outcomes and if yes, which factors had the strongest association. This has been explained with the reason/justification in the last paragraph of the introduction. See page 6, line 111-115.

However, this did not affect any initial methods related to data collection reported in the original protocol. We did not collect any additional data to respond to the objectives of this manuscript. 

As stated in the original protocol, we have also included in this revised version the prevalence of hypertension in pregnancy among hypocalcaemic women. See data analysis section, page 11, line 194-196 and results section, page 13, line 233-239.

3. Are the conclusions supported by the data and do they address the research question presented in the Registered Report Protocol?

The manuscript must describe a technically sound piece of scientific research with data that supports the conclusions. The conclusions must be drawn appropriately based on the research question(s) outlined in the Registered Report Protocol and on the data presented.

Reviewer #1: No

Reviewer #2: Yes

 Resp: Thank you for your answer. The conclusions of this manuscript respond to objectives of the original research protocol. These objectives were just adapted to make more meaning. The conclusions of this manuscript are based on the objectives of the manuscript.

4. Have the authors made all data underlying the findings in their manuscript fully available?

Reviewer #1: No

Reviewer #2: Yes

 Resp: Thanks for your answer. We made all data underlying the findings in this manuscript fully available by submitting the data base with the previous revision.

5. Is the manuscript presented in an intelligible fashion and written in standard English?

Reviewer #1: No

Reviewer #2: Yes

 Resp: Thanks for your answer. The manuscript has been read and edited by a native English speaker.

Review comments to the authors

Specific Reviewer 1 comments

Comment: An article seems more sophisticated and scientific.

However, its devised version does not change the results and their clinical meanings to improve outcomes on maternal and fetal health. 

Resp: Thanks very much for your observation. We hope that our explanations help clarify some issues

Comment: The authors found that ionised Ca of the pregnant mothers seems to have impacts on fetal outcomes, including Apgar scores at one and five minutes after birth. However, the issues of neonates must be occurrence of events seen at longer observational periods of time, such as 7-day, 14-day, or 28-day mortality or IRDS. If neonates have lower Apgar score (AS) at birth, neonatologists could treat them properly and AS seems no effects on 7-, 14-,28-days mortality. 

Resp: Thank you very much for your comment. Our manuscript presents results from a registered protocol. We perfectly agree that neonatal follow up periods extend to 28days, and our study did not look as far as that to see if other low calcium-related adverse outcomes could occur. So long term effects of low calcium were not our target (this has been included as a limit of this study). See page 25, line 421-424.

According our methods, the factors (were low calcium, low albumin and calcium supplementation in the mother), and our outcomes of interest were measured at term and birth. All selected foetal outcomes have been shown in literature to have potential negative effects on the health of the child in future. For example, low Apgar scores have been associated with poor neurological development and lower IQ (2,3), respiratory distress, hypothermia, feeding problems (4), higher neonatal mortality (5). Similar effects can be cited for low birth weight and others.

More interest is put on which maternal variables could be acted upon to reduce adverse maternal and foetal outcomes. We found out that the key determinant was the ionised calcaemic state, sometimes the calcium supplementation. 

Children with poor Apgar scores could be resuscitated by neonatologist as stated by the reviewer and the course of evolution might not show any long term effects. But we feel that detecting a factor like the calcaemic state of the mother (and acting on it) could help reduce the number of children born with low Apgar score. The same goes for children with low birth weight (responsible for 60-80%) of neonatal deaths (6). Preventing the birth of babies with low Apgar scores and low birthweight is, therefore, better than having neonatologists ready to resuscitate these babies.

Moreover, our finding of the effect of ionised calcaemic state on hypertensive disease in pregnancy is a turning point. Acting only at this level prevents its occurrence and adverse maternal and foetal outcomes associated with hypertension in pregnancy (especially preeclampsia and eclampsia).

Comment: In conclusion, although the authors had tremendously a lot of time to study, the results seems no clinical impacts on neonates and this might be rejected. 

 Resp: Thanks for your comment. We hope that our explanations have convincing enough. Do maternal variables like calcaemic state, albuminaemia state and calcium supplementation affect maternal and foetal outcomes? Can these factors be acted upon to prevent the occurrence of adverse outcomes? We think the answer is yes. In this light, we believe our findings can contribute to improving maternal and feotal health.

Reviewer 2 comments

All comments raised by the first and second revision are addressed:

Thanks for that and congratulation. 

Resp: Thank you very much for your valuable contributions

 

References

1. Ajong AB, Kenfack B, Ali IM, Yakum MN, Aljerf L, Telefo PB. Hypocalcaemia and calcium intake in pregnancy: A research protocol for critical analysis of risk factors, maternofoetal outcomes and evaluation of diagnostic methods in a third-category health facility, Cameroon. PLoS One. 2020;15(11 November). 

2. Hassen TA, Chojenta C, Egan N, Loxton D. The association between the five-minute apgar score and neurodevelopmental outcomes among children aged 8−66 months in Australia. Int J Environ Res Public Health. 2021;18(12). 

3. Tweed EJ, Mackay DF, Nelson SM, Cooper SA, Pell JP. Five-minute Apgar score and educational outcomes: Retrospective cohort study of 751 369 children. Arch Dis Child Fetal Neonatal Ed. 2016;101(2):F121–6. 

4. Thavarajah H, Flatley C, Kumar S. The relationship between the five minute Apgar score, mode of birth and neonatal outcomes. J Matern Neonatal Med. 2018;31(10):1335–41. 

5. Cnattingius S, Norman M, Granath F, Petersson G, Stephansson O, Frisell T. Apgar Score Components at 5 Minutes: Risks and Prediction of Neonatal Mortality. Paediatr Perinat Epidemiol. 2017;31(4):328–37. 

6. Njim T, Atashili J, Mbu R, Choukem SP. Low birth weight in a sub-urban area of Cameroon: An analysis of the clinical cut-off, incidence, predictors and complications. BMC Pregnancy Childbirth. 2015;15(1).

---

## [Decision Letter · Decision Letter 2]

2 May 2022

PONE-D-21-33966R2

How do ionised calcium, total calcium, total albumin, and calcium supplementation affect maternofoetal outcomes? analysis from a resource-limited setting

PLOS ONE

Dear Dr. Ajong,

Thank you for submitting your manuscript to PLOS ONE. After careful consideration, we have decided that your manuscript does not meet our criteria for publication and must therefore be rejected.

I am sorry that we cannot be more positive on this occasion, but hope that you appreciate the reasons for this decision.

Kind regards,

Linglin Xie

Academic Editor

PLOS ONE

Reviewers' comments:

Reviewer's Responses to Questions

**Comments to the Author**

1. Does the manuscript adhere to the experimental procedures and analyses described in the Registered Report Protocol?

If the manuscript reports any deviations from the planned experimental procedures and analyses, those must be reasonable and adequately justified.

Reviewer #1: Yes

2. If the manuscript reports exploratory analyses or experimental procedures not outlined in the original Registered Report Protocol, are these reasonable, justified and methodologically sound?

A Registered Report may include valid exploratory analyses not previously outlined in the Registered Report Protocol, as long as they are described as such.

Reviewer #1: Yes

3. Are the conclusions supported by the data and do they address the research question presented in the Registered Report Protocol?

The manuscript must describe a technically sound piece of scientific research with data that supports the conclusions. The conclusions must be drawn appropriately based on the research question(s) outlined in the Registered Report Protocol and on the data presented.

Reviewer #1: No

4. Have the authors made all data underlying the findings in their manuscript fully available?

Reviewer #1: No

5. Is the manuscript presented in an intelligible fashion and written in standard English?

Reviewer #1: No

6. Review Comments to the Author

Please use the space provided to explain your answers to the questions above. (Please upload your review as an attachment if it exceeds 20,000 characters)

Reviewer #1: Reviewer’s comment on PONE-D-21-33966-R2

Major comments

This is revised article studying an association between adverse events of neonate and biochemical parameters of serum calcium and albumin concentration of pregnant mothers at 37 and over of pregnant period before delivering neonates. The authors found associations of maternal lower ionized calcium level <= 131 mmol/L and low birth weight, low birth length, low brachial circumference at birth, and low Apgar score at one and fifth minutes after birth. Moreover, they stated that calcium supplementation significantly improved the Apgar scores at the first and fifth minutes showing their results in Table 3. Form reading carefully their descriptions about evaluating maternal calcium supplementations during pregnant periods, it is unclear how much thy took calcium supplementation adding on dietary intake. The methodological problem that I am feeling is that daily total calcium intakes at late pregnant periods of meal and calcium supplementation must be calculated. Otherwise, it must be unclear which pregnant mother must be target to supplement calcium because she is deficient calcium intake. In addition, serum vitamin D concentrations and time durations of sunlight exposure also must be considered when which pregnant women is the target of calcium supplementations to prevent poor maternal or neonatal outcomes. Without disclosed above mentioned problems must be resolved, this article could not provide meaningful information to health care professionals.

In conclusion, I would say that this revised article must not be accepted.

7. PLOS authors have the option to publish the peer review history of their article (what does this mean?). If published, this will include your full peer review and any attached files.

Reviewer #1: **Yes: **Teruyoshi Amagai, MD, PhD, University of Jikei Health Care Sciences, Osaka, Japan

- - - - -

---

## [Author Response · Author response to Decision Letter 2]

12 May 2022

Response to Reviewers

Comment: This is revised article studying an association between adverse events of neonate and biochemical parameters of serum calcium and albumin concentration of pregnant mothers at 37 and over of pregnant period before delivering neonates. The authors found associations of maternal lower ionized calcium level <= 131 mmol/L and low birth weight, low birth length, low brachial circumference at birth, and low Apgar score at one and fifth minutes after birth. Moreover, they stated that calcium supplementation significantly improved the Apgar scores at the first and fifth minutes showing their results in Table 3. 

Response: Thank you for the review. Our article evaluates associations between hypocalcalcaemic states and maternofoetal outcomes. We have reviewed the title to fit the design of the study. In essence, our study evaluated associations which could be potential hypotheses for future longitudinal studies (for cause-effect relationships). This is recognized as a limit to our findings in the manuscript. See title and section on limits.

Comment: Form reading carefully their descriptions about evaluating maternal calcium supplementations during pregnant periods, it is unclear how much thy took calcium supplementation adding on dietary intake. The methodological problem that I am feeling is that daily total calcium intakes at late pregnant periods of meal and calcium supplementation must be calculated. Otherwise, it must be unclear which pregnant mother must be target to supplement calcium because she is deficient calcium intake. In addition, serum vitamin D concentrations and time durations of sunlight exposure also must be considered when which pregnant women is the target of calcium supplementations to prevent poor maternal or neonatal outcomes. Without disclosed above mentioned problems must be resolved, this article could not provide meaningful information to health care professionals.

In conclusion, I would say that this revised article must not be accepted. 

Response: Thank you for your concern. Our objective in this manuscript was not to evaluate the calcium intake of the participants in the diet and supplements. As stated in response to reviewers 1, We have a manuscript evaluating calcium intake (in terms of calcium supplementation practices in pregnancy) and determines factors affecting this practice in pregnancy (given that it is recommended by the WHO to be systematic in LMIC for the prevention of hypertensive diseases in pregnancy). This was submitted on the 1st 0ct 2021 (PONE-D-21-31679). 

Our key problem was to evaluate the relative strength of association that ionized and total calcaemic states could have on maternofoetal outcomes. In doing this we attached a simple question to see if there was an association between “haven taken calcium supplements in pregnancy” and maternofoetal outcomes. We were not looking for which women to supplement and which women not to supplement. 

We understand that Vit D concentrations and exposure to sunlight contribute to providing active calcitriol for intestinal absorption of calcium. Vit D and sunlight exposure concur to increasing concentrations of calcitriol which increases intestinal calcium absorption. Therefore, we believe that to evaluate the association between calcaemic states and maternofoetal outcomes, there is no need to control for sunlight exposure and Vit D concentrations. This because the terminal point of Vit D concentrations and sunlight exposure is calcaemia.

---

## [Editor Report · Decision Letter 3]

5 Jul 2022

Adverse maternofoetal outcomes associated with ionised calcaemia, total calcaemia, albuminaemia, and calcium supplementation in pregnancy: analysis from a resource-limited setting

PONE-D-21-33966R3

Dear Dr. Ajong,

We’re pleased to inform you that your manuscript has been judged scientifically suitable for publication and will be formally accepted for publication once it meets all outstanding technical requirements.

Kind regards,

Dylan A Mordaunt, MD, MPH, FRACP

Academic Editor

PLOS ONE

Additional Editor Comments (optional):

Thank you for your submission. I have taken over as academic editor and have familiarised myself with the previous reviews and decisions. Fundamentally, the recommendations have been addressed by the reviewers and the manuscript now meets PLoS One's criteria for publication. Some of the somewhat unclear comments from the reviews do not directly relate to PLoS One's criteria for publication, and the manuscript reflects execution of a peer reviewed protocol.

With specific reference to the criteria for publication:

1. The study appears to present the results of original research. Reviewer comments seem to address the perceived value of the research and a fixed perspective on limitations rather than interpreting the data in the context of an a priori agreed protocol.

2. Results reported do not appear to have been published elsewhere.

3. Experiments, statistics, and other analyses are performed to a high technical standard and are described in sufficient detail. Again, review comments do not detract from this and indeed the protocol was peer reviewed and published in PLoS One. The stated limitations are a lack of presentation of data not collected. Momentary ionized calcium is not primarily a reflection of dietary calcium, so I'm not convinced this is a critical limitation, and wasn't found to be in the peer-reviewed protocol.

4. Conclusions are presented in an appropriate fashion and are supported by the data.

5. The article is presented in an intelligible fashion and is written in standard English. There are some "turns of phrase" or possibly idioms that I wasn't familiar with, such as "inexistent", but these appear to be accepted synonyms and reflect international dialects of English rather than poor fluency.

6. The research meets all applicable standards for the ethics of experimentation and research integrity.

7. The article adheres to appropriate reporting guidelines and community standards for data availability. A report like this could benefit from reference to a standardised reporting checklist like STROBE or extensions, but the way this is structured is high quality and all the detail is there, it would be a minor value add and optional for the authors.
---

## [Editor Report · Acceptance letter]

21 Jul 2022

PONE-D-21-33966R3 

Adverse maternofoetal outcomes associated with ionised calcaemia, total calcaemia, albuminaemia, and calcium supplementation in pregnancy: analysis from a resource-limited setting 

Dear Dr. Ajong:

I'm pleased to inform you that your manuscript has been deemed suitable for publication in PLOS ONE. Congratulations! Your manuscript is now with our production department. 

Kind regards, 

on behalf of

Associate Professor Dylan A Mordaunt 

Academic Editor

PLOS ONE